# Pseudo-Differential Neural Operator: Generalized Fourier Neural Operator for Learning Solution Operators of Partial Differential Equations

**Jin Young Shin**[*]                                                      *sjy6006@postech.ac.kr*
*Department of Mathematics*
*Pohang University of Science and Technology (POSTECH), Pohang 37673, Republic of Korea.*

**Jae Yong Lee**[*]                                                        *jaeyong@cau.ac.kr*
*Artificial Intelligence Graduate School*
*Chung-Ang University, Seoul 06974, Republic of Korea.*

**Hyung Ju Hwang**[†]                                                      *hjhwang@postech.ac.kr*
*Department of Mathematics*
*Pohang University of Science and Technology (POSTECH), Pohang 37673, Republic of Korea.*

**Reviewed on OpenReview:** *https://openreview.net/forum?id=8O5jKZOGqf*

## Abstract

Learning the mapping between two function spaces has garnered considerable research attention. However, learning the solution operator of partial differential equations (PDEs) remains a challenge in scientific computing. Fourier neural operator (FNO) was recently proposed to learn solution operators, and it achieved an excellent performance. In this study, we propose a novel *pseudo-differential integral operator* (PDIO) to analyze and generalize the Fourier integral operator in FNO. PDIO is inspired by a pseudo-differential operator, which is a generalized differential operator characterized by a certain symbol. We parameterize this symbol using a neural network and demonstrate that the neural network-based symbol is contained in a smooth symbol class. Subsequently, we verify that the PDIO is a bounded linear operator, and thus is continuous in the Sobolev space. We combine the PDIO with the neural operator to develop a *pseudo-differential neural operator* (PDNO) and learn the nonlinear solution operator of PDEs. We experimentally validate the effectiveness of the proposed model by utilizing Darcy flow and the Navier-Stokes equation. The obtained results indicate that the proposed PDNO outperforms the existing neural operator approaches in most experiments.

## 1 Introduction

In science and engineering, several physical systems and natural phenomena are described by partial differential equations (PDEs) (Courant & Hilbert, 1953). Approximating the solution of the underlying PDEs is crucial to understanding and predicting a system. Conventional numerical methods, such as finite difference methods (FDMs) and finite element methods, involve a trade-off between accuracy and the time required. In several complex systems, it may be excessively time-consuming to employ numerical methods to obtain accurate solutions. Furthermore, in some cases, the underlying PDE may be unknown.

With remarkable advancements in deep learning, several studies have focused on utilizing deep learning to solve PDEs (Nabian & Meidani, 2018; E & Yu, 2018; Sirignano & Spiliopoulos, 2018; Raissi et al., 2019;

---

[*]Equal contribution.
[†]Corresponding author.

Hwang et al., 2020; Lee et al., 2021). An example is an operator learning (Guo et al., 2016; Bhatnagar et al., 2019; Khoo et al., 2021), which utilizes neural networks to parameterize mapping from the parameters (external force, initial, and boundary condition) of the given PDE to the solutions of that PDE. In addition, several studies have employed different convolutional neural networks as surrogate models to solve various problems, such as the uncertainty quantification tasks for PDEs (Zhu & Zabaras, 2018; Zhu et al., 2019) and PDE-constrained control problems (Holl et al., 2020; Hwang et al., 2021). Based on the universal approximation theorem of the operator (Chen & Chen, 1995), *DeepONet* was introduced by Lu et al. (2019). In follow-up studies, extension models of the DeepONet were proposed (Wang et al., 2021; Kissas et al., 2022; Lee et al., 2023).

Another approach to operator learning is via *neural operator*, proposed by Li et al. (2020c;b;a). Li et al. (2020c) proposed an iterative architecture inspired by Green's function of elliptic PDEs. The iterative architecture comprises a linear transformation, an integral operator, and a nonlinear activation function, allowing the architecture to approximate complex nonlinear mapping. As an extension of this research, Li et al. (2020b) adopted a multi-pole method to develop a multi-scale graph structure. Gupta et al. (2021) approximated the kernel of the integral operator using the multiwavelet transform. Recently, various operator learning models based on neural operators have emerged, such as those presented by Tripura & Chakraborty (2023) and Rahman et al. (2022). The review papers (Lu et al., 2022; Goswami et al., 2023), along with their references, encompass a diverse range of operator learning models.

As one of such models, Li et al. (2020a) proposed a *Fourier integral operator* using fast Fourier transform (FFT) to minimize the cost of approximating the integral operator. They directly parameterized the kernel in the Fourier integral operator by its Fourier space coefficients, which only depend on frequency mode. Here, we analyze the Fourier integral operator from the perspective of *pseudo-differential operators* (PDOs). PDOs are generalized linear partial differential operators and have been extensively studied mathematically (Boutet de Monvel, 1971; Hörmander, 2007; Ruzhansky & Turunen, 2009; Taylor, 2017). A *pseudo-differential integral operator* (PDIO) is proposed to generalize the Fourier integral operator in the FNO based on the PDO. A neural network called a symbol network is utilized to approximate the PDO symbols [1] The proposed symbol network is contained in a toroidal class of symbols; hence, a PDIO is a continuous operator in the Sobolev space. Furthermore, the PDIO can be applied to the solution operator of time-dependent PDEs using a time-dependent symbol network.

The main contributions of this study are as follows.

- The Fourier integral operator proposed in Li et al. (2020a) is interpreted from a PDO perspective. The symbol of the Fourier integral operator only depends on frequency domain $\xi$, rather than position $x$. Furthermore, the symbol may not be contained in a toroidal symbol class; hence, the Fourier integral operator cannot be guaranteed to be a continuous operator.

- A novel PDIO is proposed based on the PDO to generalize the Fourier integral operator. PDIO approximates the PDO using symbol networks. We demonstrate that the proposed symbol network is contained in a toroidal symbol class of PDOs, thus implying that the PDIO with the symbol network is a continuous operator in the Sobolev space.

- Time-dependent PDIO, a PDIO with time-dependent symbol networks, can be utilized to approximate the solution operator of time-dependent PDEs. It approximates the solution operator, including the solution for time $t$, which is not in the training data. Furthermore, it is a continuous-in-time operator.

- A *pseudo-differential neural operator* (PDNO), which comprises a linear combination of our PDIOs and the neural operator proposed in Li et al. (2020c), is developed. PDNO outperforms the existing operator learning models, such as the FNO by Li et al. (2020a) and the multiwavelet-based operator by Gupta et al. (2021), in hard problems (Darcy flow and Navier-Stokes equation). In particular, the PDNO reduces overfitting better than other models (Figure 1).

---

[1]Guibas et al. (2021) proposed *adaptive FNO* (AFNO), which adopts a single layer neural network in the Fourier domain. Our application of a neural network differs from AFNO. Although we utilized neural networks to parameterize the PDO symbol, AFNO adopts a neural network that directly takes frequency variable as input (Refer to Figure 2).

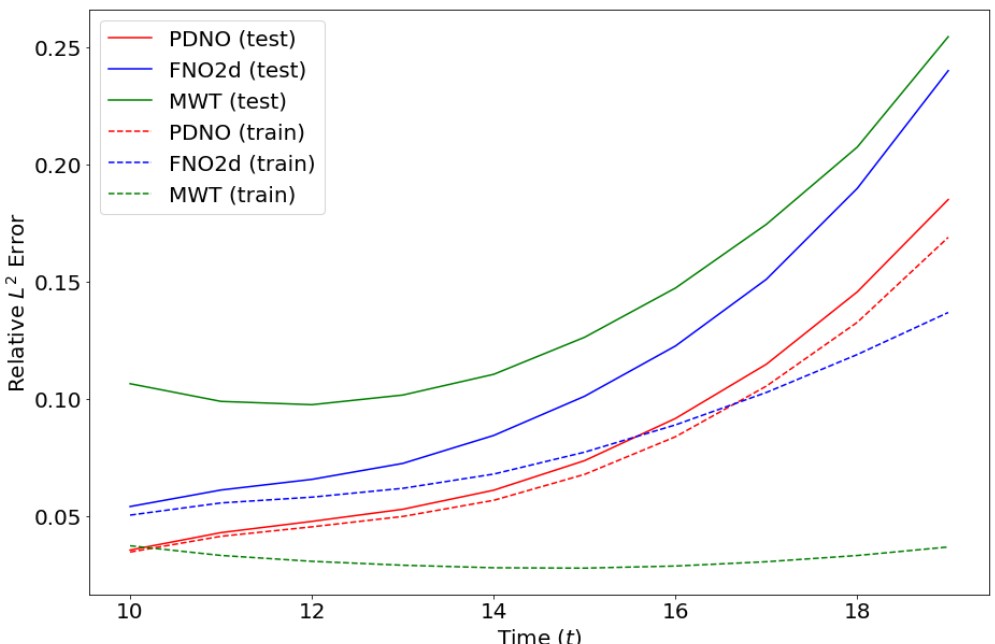

Figure 1: Comparison of the train and the test relative $L^2$ error by time horizon $t = 10, ..., 19$ on the Navier-Stokes equation with viscosity $\nu = 10^{-5}$. FNO and MWT are highly overfitted, while PDNO is not. See Section 5.2 for detailed experimental setups regarding the Navier-Stokes equation.

## 2 Fourier integral operator and PDO

Here, we attempt to approximate an operator $\mathcal{G} : \mathcal{A} \rightarrow \mathcal{U}$ between two function spaces $\mathcal{A}$ and $\mathcal{U}$. The operator $\mathcal{G}$ can be considered as the solution operators of various parametric PDEs (kindly refer to Section 5 for examples). To determine a map from the function $f(x) \in \mathcal{A}$ to the solution $u(x) \in \mathcal{U}$, we introduce a neural operator architecture to effectively learn infinite-dimensional operators.

### 2.1 Neural operator

Inspired by Green's functions of elliptic PDEs, Li et al. (2020c) proposed an iterative *neural operator* to approximate the solution operators of parametric PDEs. First, the input $f(x)$ was lifted to a higher representation $f_0(x) = P(f(x))$. Next, the iterations $f_0 \mapsto f_1 .... \mapsto f_T$ were applied using the update $f_t \mapsto f_{t+1}$ formulated by Eq.1::

$$f_{t+1}(x) = \sigma \left( W f_t(x) + \mathcal{K}_\phi[f_t](x) \right), \tag{1}$$

for $t = 0, ..., T - 1$, where $W$ is a local linear transformation and $\sigma$ is a nonlinear activation function. $\mathcal{K}_\phi$ denotes an integral operator $\mathcal{K}$ parameterized by $\phi$ and expressed as

$$\mathcal{K}_\phi[f_t](x) = \int_D \kappa_\phi(x, y) f_t(y) dy, \tag{2}$$

where $D$ represents a bounded domain of the input function. The output $u(x) = Q(f_T(x))$ is the projection of $f_T(x)$ by the local transformation $Q$. Several studies have considered the best approach to choosing the kernel function $\kappa_\phi$ and computing the corresponding integral operator. The integral operator $\mathcal{K}_\phi$ can be parameterized using message passing on graph networks (Li et al., 2020c). Here, we focused on the *Fourier integral operator* proposed by Li et al. (2020a).

## 2.2 Fourier integral operator

Li et al. (2020a) proposed a neural operator structure with the integral operator $\mathcal{K}_R$ called Fourier integral operator. By letting $\kappa(x, y) = \kappa(x - y)$ and utilizing the convolution theorem, they define the Fourier integral operator as

$$\mathcal{K}_R[f_t](x) = \mathcal{F}^{-1}\left[\mathcal{F}[\kappa] \cdot \mathcal{F}[f_t](\xi)\right](x) = \mathcal{F}^{-1}\left[R(\xi) \cdot \mathcal{F}[f_t](\xi)\right](x), \tag{3}$$

where $\mathcal{F}$ denotes the Fourier transform and $\mathcal{F}^{-1}$ is its inverse. Note that the parameter $R(\xi)$ is directly parameterized on the discrete space $\xi \in \mathbb{Z}^n$. The Fourier integral operator can be extended as the generalization concept of a differential operator, PDO.

## 2.3 PDO

PDOs have been studied since the 1960's. We consider a PDE $\mathcal{L}_a[u(x)] = f(x)$ with a linear differential operator $\mathcal{L}_a = \sum_\alpha c_\alpha D^\alpha$. To determine a map $T$ from $f$ to $u$, we apply the Fourier transform to obtain the following:

$$a(\xi)\hat{u} \stackrel{\text{def}}{=} \left(\sum_\alpha c_\alpha(i\xi)^\alpha\right)\hat{u} = \hat{f}, \tag{4}$$

where $\xi \in \mathbb{R}^n$ represents variables in the Fourier space and $\hat{f}(\xi)$ denotes a Fourier transform of function $f(x)$. If $a(\xi)$ never attains zero, we obtain the solution operator of the PDE as

$$u(x) = T(f)(x) \stackrel{\text{def}}{=} \int_{\mathbb{R}^n} \frac{1}{a(\xi)}\hat{f}(\xi)e^{2\pi i\xi x}d\xi. \tag{5}$$

A PDO can be defined as a generalization of differential operators by replacing $\frac{1}{a(\xi)}$ with $a(x, \xi)$, called a symbol (Hörmander, 2003; 2007). First, we define a symbol $a(x, \xi)$ and a class of symbols.

**Definition 1.** *Let $0 < \rho \leq 1$ and $0 \leq \delta < 1$. A function $a(x, \xi)$ is called a Euclidean symbol on $\mathbb{T}^n \times \mathbb{R}^n$ in a class $S_{\rho,\delta}^m(\mathbb{T}^n \times \mathbb{R}^n)$, where $\mathbb{T}^n$ is n-dimensional torus if $a(x, \xi)$ is smooth on $\mathbb{T}^n \times \mathbb{R}^n$ and satisfies the following inequality:*

$$|\partial_x^\beta \partial_\xi^\alpha a(x, \xi)| \leq c_{\alpha\beta}\langle\xi\rangle^{m-\rho|\alpha|+\delta|\beta|}, \tag{6}$$

*for all $\alpha, \beta \in \mathbb{N}_0^n$, and for all $x \in \mathbb{T}^n$ and $\xi \in \mathbb{R}^n$, where a constant $c_{\alpha\beta}$ may depend on $\alpha$ and $\beta$ but not on $x$ and $\xi$. Here, $\langle\xi\rangle \stackrel{\text{def}}{=} (1 + \|\xi\|^2)^{1/2}$ with the Euclidean norm $\|\xi\|$.*

Note that $m$, $\rho$, and $\delta$ are related to the regularity of the symbol. The PDO corresponding to the symbol class $S_{\rho,\delta}^m(\mathbb{T}^n \times \mathbb{R}^n)$ can be defined as follows.

**Definition 2.** *The Euclidean PDO $T_a : \mathcal{A} \to \mathcal{U}$ with the Euclidean symbol $a(x, \xi) \in S_{\rho,\delta}^m(\mathbb{T}^n \times \mathbb{R}^n)$ is defined as*

$$T_a(f)(x) = \int_{\mathbb{R}^n} a(x, \xi)\hat{f}(\xi)e^{2\pi i\xi x}d\xi, \tag{7}$$

*where $\hat{f}(\xi)$ denotes the Fourier transform of function $f(x)$.*

The Euclidean PDO can be re-written using the Fourier transform as

$$T_a(f)(x) = \mathcal{F}^{-1}\left[a(x, \xi)\mathcal{F}[f](\xi)\right]. \tag{8}$$

## 2.4 Difference between Fourier integral operator and PDO

Comparing the Fourier integral operator $\mathcal{K}_R$ in (3) and Euclidean PDO $T_a$ in (8), there are two main differences. There is a question about whether it is appropriate to parameterize $R(\xi)$ directly on the discrete space $\xi \in \mathbb{Z}^n$ without treating $\xi$ as a continuous variable. Furthermore, parameters $R(\xi)$ only consider the dependency on $\xi$, while the symbol $a(x, \xi)$ has a dependency on $x$. To generalize the Fourier integral operator based on PDO, the key idea of our method is to parameterize the Euclidean symbol using neural networks to

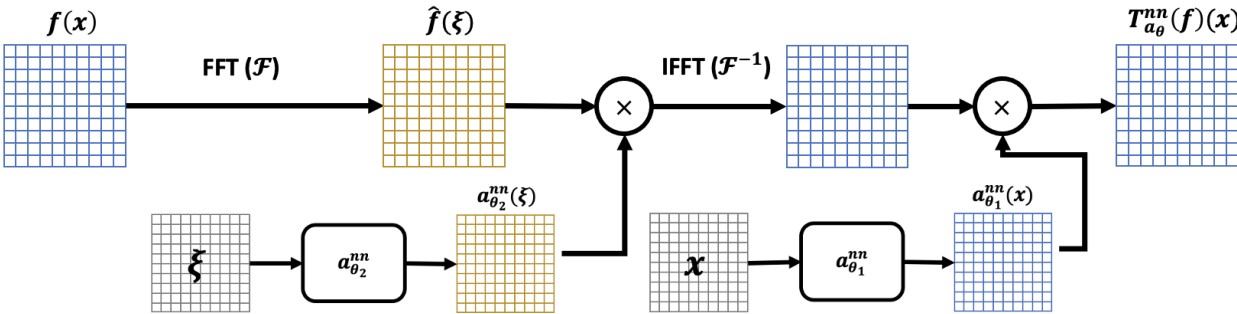

Figure 2: An architecture of a PDIO with symbol networks $a_{\theta_1}^{nn}(x)$ and $a_{\theta_2}^{nn}(\xi)$. Considering that FFT and inverse FFT are used, both the input and output are in the form of uniform mesh. Each value $a_{\theta_1}^{nn}(x)$ and $a_{\theta_2}^{nn}(\xi)$ is obtained from separate neural networks.

render the symbol smooth. This makes the model smooth, thereby mitigating overfitting by smoothening the symbols (Figure 1 and Proposition 1).

The following section introduces the PDO theory and proposed model based on the PDO. We also derive the proposed model's smoothness from the neural network's smoothness.

## 3   Proposed integral operator : PDIO

In this section, we cover three parts. First, we discuss the relationship between the symbols and smoothness of PDOs that have been studied in the past. Next, we turn to the relationship between FNOs and PDOs. It is important to note that, although FNOs can be considered discrete versions of PDOs, they do not possess sufficiently smooth symbols. In the final part, we propose a PDIO model that has a sufficiently smooth symbol. We present concrete evidence that a neural network using a GELU-like activation function can generate such a smooth symbol (Appendix E).

### 3.1   Symbol network and PDIO

The primary idea in our study is to parameterize the Euclidean symbol $a(x, \xi)$ using neural networks $a_\theta^{nn}(x, \xi)$. This network is called a *symbol network*. The symbol network $a_\theta^{nn}(x, \xi)$ is assumed to be factorized into $a_\theta^{nn}(x, \xi) = a_{\theta_1}^{nn}(x)a_{\theta_2}^{nn}(\xi)$ (refer to B.1). Both smooth functions $a_{\theta_1}^{nn}(x)$ and $a_{\theta_2}^{nn}(\xi)$ are parameterized by fully connected neural networks. We propose a PDIO to approximate the Euclidean PDO using the symbol network and Fourier transform as follows:

$$\mathcal{K}_a[f](x) := \mathcal{F}^{-1}\left[a_\theta^{nn}(x, \xi)\mathcal{F}[f](\xi)\right] = a_{\theta_1}^{nn}(x)\mathcal{F}^{-1}\left[a_{\theta_2}^{nn}(\xi)\mathcal{F}[f](\xi)\right], \tag{9}$$

where $\mathcal{F}$ denotes the Fourier transform and $\mathcal{F}^{-1}$ is its inverse. The diagram of the PDIO is presented in Figure 2.

Practically, $\mathcal{F}$ and $\mathcal{F}^{-1}$ in (9) are approximated by the FFT, which is an effective algorithm that computes the discrete Fourier transform (DFT). Although the symbol network $a_{\theta_2}^{nn}(\xi)$ is defined on $\mathbb{R}^n$, the inverse DFT is evaluated only on the discrete space $\mathbb{Z}^n$. Therefore, the symbol network $a_\theta^{nn}(x, \xi)$ should be considered on the restricted domain $\mathbb{T}^n \times \mathbb{Z}^n$ (kindly refer to B.2 for details). The following section details the definitions and properties of the symbol and PDO on $\mathbb{T}^n \times \mathbb{Z}^n$ to elucidate the PDIO on the domain $\mathbb{T}^n \times \mathbb{Z}^n$. Moreover, we introduce a theorem that bridges the gap between the Euclidean symbol and the restricted Euclidean symbol.

### 3.2   PDOs on $\mathbb{T}^n \times \mathbb{Z}^n$

The discretization of the Euclidean symbol and Euclidean PDO are defined in this section.

**Definition 3.** *A toroidal symbol class is a set $S_{\rho,\delta}^m(\mathbb{T}^n \times \mathbb{Z}^n)$ comprising toroidal symbols $a(x,\xi)$, which are smooth in $x$ for all $\xi \in \mathbb{Z}^n$, and satisfy the following inequality:*

$$| \triangle_\xi^\alpha \, \partial_x^\beta a(x,\xi)| \leq c_{\alpha\beta} \langle \xi \rangle^{m-\rho|\alpha|+\delta|\beta|}, \tag{10}$$

*for all $\alpha, \beta \in \mathbb{N}_0^n$, and for all $(x,\xi) \in \mathbb{T}^n \times \mathbb{Z}^n$. Here, $\triangle_\xi^\alpha$ represents the difference operators.*

The PDO corresponding to the symbol class $S_{\rho,\delta}^m(\mathbb{T}^n \times \mathbb{Z}^n)$ can be defined as follows:

**Definition 4.** *The toroidal PDO $T_a : \mathcal{A} \to \mathcal{U}$ with the toroidal symbol $a(x,\xi) \in S_{\rho,\delta}^m(\mathbb{T}^n \times \mathbb{Z}^n)$ is defined by the following equation:*

$$T_a(f)(x) = \sum_{\xi \in \mathbb{Z}^n} a(x,\xi) \hat{f}(\xi) e^{2\pi i \xi x}. \tag{11}$$

It is well-known that the toroidal PDO $T_a(f)$ with $f \in C^\infty(\mathbb{T}^n)$ is well defined and $T_a(f) \in C^\infty(\mathbb{T}^n)$ (Ruzhansky & Turunen, 2009).

Here, it is necessary to prove that the restricted symbol network $a_\theta^{nn}|_{\mathbb{T}^n \times \mathbb{Z}^n}$ belongs to a certain toroidal symbol class. To connect the symbol network $a_\theta^{nn}$ and restricted symbol network $a_\theta^{nn}|_{\mathbb{T}^n \times \mathbb{Z}^n}$, we introduce an appropriate theorem that connects the symbols between the Euclidean and toroidal symbols.

**Theorem 1.** *(Ruzhansky & Turunen, 2009)* ***(Connection between two symbols)*** *Let $0 < \rho \leq 1$ and $0 \leq \delta \leq 1$. A symbol $\tilde{a} \in S_{\rho,\delta}^m(\mathbb{T}^n \times \mathbb{Z}^n)$ is a toroidal symbol if and only if there exists a Euclidean symbol $a \in S_{\rho,\delta}^m(\mathbb{T}^n \times \mathbb{R}^n)$ such that $\tilde{a} = a|_{\mathbb{T}^n \times \mathbb{Z}^n}$.*

Therefore, it is sufficient to consider whether the symbol network $a_\theta^{nn}(x,\xi)$ belongs to a certain Euclidean symbol class.

## 3.3 Propositions on the symbol network and PDIO

We demonstrate that the symbol network $a_\theta^{nn}(x,\xi)$ with the Gaussian error linear unit (GELU) activation function Hendrycks & Gimpel (2016) is contained in a certain Euclidean symbol class using the following proposition:

**Proposition 1.** *Suppose the symbol networks $a_{\theta_1}^{nn}(x)$ and $a_{\theta_2}^{nn}(\xi)$ are fully connected neural networks with nonlinear activation GELU. Then, the symbol network $a_\theta^{nn}(x,\xi) = a_{\theta_1}^{nn}(x) a_{\theta_2}^{nn}(\xi)$ is in $S_{1,0}^1(\mathbb{T}^n \times \mathbb{R}^n)$. Therefore, the restricted symbol network $\widetilde{a_\theta^{nn}} \overset{def}{=} a_\theta^{nn}|_{\mathbb{T}^n \times \mathbb{Z}^n}$ is in a toroidal symbol class $S_{1,0}^1(\mathbb{T}^n \times \mathbb{Z}^n)$.*

**Remark 1.** *Here, we focus on the most important case where $\rho = 1$ and $\delta = 0$, because $S_{\rho,\delta}^m \supset S_{1,0}^m$ for $0 < \rho \leq 1$ and $0 \leq \delta < 1$ (Hörmander, 2007). Although the proposition only considers the symbol network with GELU, it can be verified for various activation functions (refer to E).*

*Proof.* The fully connected neural network for the symbol network $a_{\theta_1}^{nn}(x)$ is denoted as follows:

$$Z_1^{[l]} = W_1^{[l]} A_1^{[l-1]} + b_1^{[l]} \quad (l = 1, 2, ..., L_1), \quad A_1^{[l]} = \sigma(Z_1^{[l]}) \quad (l = 1, 2, ..., L_1 - 1),$$

where $W_1^{[l]}$ is a weight matrix, $b_1^{[l]}$ denotes a bias vector in the $l$-th layer of the network, $\sigma$ represents an element-wise activation function, $A_1^{[0]} = x$ is an input feature vector, and $Z_1^{[L_1]} = a_{\theta_1}^{nn}(x)$ denotes an output of the network with $\theta_1 = \{W_1^{[l]}, b_1^{[l]}\}_{l=1}^{L_1}$. Similarly, we define $W_2^{[l]}$, $b_2^{[l]}$, $Z_2^{[l]}$ and $A_2^{[l]}$ $(l = 1, 2, ..., L_2)$ for the neural network $a_{\theta_2}^{nn}(\xi)$.

The neural network $a_{\theta_1}^{nn}(x)$ and its derivative are continuous on a compact set $\mathbb{T}^n$. Therefore, $|\partial_x^\beta a_{\theta_1}^{nn}(x)| \leq c_\beta$ for some constant $c_\beta > 0$ and for all $\beta \in \mathbb{N}_0^n$. For the case $|\alpha| = 0$,

$$|\partial_\xi^\alpha a_{\theta_2}^{nn}(\xi)| = |a_{\theta_2}^{nn}(\xi)| = |W_2^{[L_2]} \sigma(Z_2^{[L_2-1]}) + b_2^{[L_2]}| \leq c_\alpha \langle \xi \rangle, \tag{12}$$

for some constant $c_\alpha > 0$ because the absolute value of GELU $\sigma(z)$ is bounded by a linear function $|z|$. Notably,

$$\partial_\xi^{e_i} a_{\theta_2}^{nn}(\xi) = W_2^{[L_2]} diag\left(\sigma'\left(Z_2^{[L_2-1]}\right)\right) \times \cdots \times W_2^{[2]} diag\left(\sigma'\left(Z_2^{[1]}\right)\right) W_2^{[1]} e_i.$$

This result implies that the multi-derivatives of symbol $\partial_\xi^\alpha a_{\theta_2}^{nn}(\xi)$ with $|\alpha| \geq 1$ comprises the product of the weight matrix and the first or higher derivatives of the activation functions. Furthermore, the derivative of GELU is bounded, and the second or higher derivatives of the function asymptotically become zero rapidly, i.e., $\sigma^{(k)} \in \mathcal{S}(\mathbb{R})$ when $k \geq 2$ (refer to Definition 5). Accordingly, we have the following inequality:

$$|\partial_\xi^\alpha a_{\theta_2}^{nn}(\xi)| \leq c_\alpha \langle \xi \rangle^{1-|\alpha|}, \tag{13}$$

for all $\alpha \in \mathbb{N}_0^n$ with $|\alpha| \geq 1$ for some positive constants $c_\alpha$. We bound the derivative of the symbol network $a_\theta^{nn}(x,\xi)$ as follows:

$$|\partial_x^\beta \partial_\xi^\alpha a(x,\xi)| = |\partial_x^\beta a_{\theta_1}^{nn}(x)||\partial_\xi^\alpha a_{\theta_2}^{nn}(\xi)| \leq \underbrace{c_\alpha c_\beta}_{=c_{\alpha\beta}} \langle \xi \rangle^{1-|\alpha|}. \tag{14}$$

Therefore, the symbol network $a_\theta^{nn}(x,\xi)$ is in $S_{1,0}^1(\mathbb{T}^n \times \mathbb{R}^n)$ as defined in Definition 1. Finally, using Theorem 1, we deduce that $\widetilde{a^{nn}} = a^{nn}|_{\mathbb{T}^n \times \mathbb{Z}^n}$ is in $S_{1,0}^1(\mathbb{T}^n \times \mathbb{Z}^n)$. □

We introduce the theorem on the boundedness of a toroidal PDO as follows:

**Theorem 2.** *(Ruzhansky & Turunen, 2009)* **(Boundedness of a toroidal PDO in the Sobolev space)** *Let $m \in \mathbb{R}$ and $k \in \mathbb{N}$, which is the smallest integer greater than $\frac{n}{2}$, and let $a : \mathbb{T}^n \times \mathbb{Z}^n \to \mathbb{C}$ such that*

$$|\partial_x^\beta \triangle_\xi^\alpha a(x,\xi)| \leq C\langle \xi \rangle^{m-|\alpha|} \quad \text{for all } (x,\xi) \in \mathbb{T}^n \times \mathbb{Z}^n, \tag{15}$$

*and all multi-indices $\alpha$ such that $|\alpha| \leq k$ and all multi-indices $\beta$. Then, the corresponding toroidal PDO $T_a$ defined in Definition 4 extends to a bounded linear operator from the Sobolev space $W^{p,s}(\mathbb{T}^n)$ to the Sobolev space $W^{p,s-m}(\mathbb{T}^n)$ for all $1 < p < \infty$ and any $s \in \mathbb{R}$.*

The restricted symbol network $\widetilde{a_\theta^{nn}}$ is in a toroidal symbol class $S_{1,0}^1(\mathbb{T}^n \times \mathbb{Z}^n)$ from Proposition 1. Hence, it satisfies the condition in Theorem 2. Therefore, the PDIO $\mathcal{K}_a$ (9) with the restricted symbol network $\widetilde{a_\theta^{nn}}$ is a bounded linear operator from $W^{p,s}(\mathbb{T}^n)$ to $W^{p,s-1}(\mathbb{T}^n)$ for all $1 < p < \infty$ and $s \in \mathbb{R}$. This implies that the PDIO is a continuous operator between the Sobolev spaces. Therefore, we expect that the PDIO can be applied to a neural operator (1) to obtain a smooth and general solution operator. A description of its application to the neural operator is presented in Section 4.1.

# 4 Neural operator with PDIOs

## 4.1 PDNO

Using the proposed integral operator (9) with the neural operator (1), the combined model is called a PDNO. Consider the general case in which the input function $f_t(x)$ and output function $f_{t+1}(x)$ are vector functions. Let $f_t(x) = [f_{t,i}(x) : \mathbb{R}^n \to \mathbb{R}]_{i=1}^{c_{in}} \in \mathbb{R}^{c_{in}}$ with the number of input channels $c_{in}$ and $x \in \mathbb{R}^n$. Then, the PDIO $\mathcal{K}_a$ is expressed as:

$$\mathcal{K}_a(f_t)(x) = \left[ \sum_{i=1}^{c_{in}} a_{\theta_1,ij}^{nn}(x) \mathcal{F}^{-1} \left[ a_{\theta_2,ij}^{nn}(\xi) \mathcal{F}[f_{t,i}](\xi) \right] (x) \right]_{j=1}^{c_{out}}, \tag{16}$$

where $\theta_1, \theta_2$ denote the parameters of each symbol network and $c_{out}$ represents the number of output channels with $f_{t+1}(x) \in \mathbb{R}^{c_{out}}$. Certainly, the symbol network has $c_{in} \times c_{out}$ outputs for each channel as illustrated in Figure 6. In the experiments, we use three separate symbol networks $a_{\theta_1}^{nn}(x)$, $\text{Re}\left(a_{\theta_2}^{nn}(\xi)\right)$, and $\text{Im}\left(a_{\theta_2}^{nn}(\xi)\right)$. Each symbol network takes $x \in \mathbb{R}^n$ and $\xi \in \mathbb{R}^n$ as input, and generate $c_{in} \times c_{out}$ values.

## 4.2 Time-dependent PDIO

Consider the time-dependent PDE

$$\frac{\partial u}{\partial t} = \mathcal{L}u, \quad u(x,0) = u_0(x), \quad (x,t) \in \mathbb{T}^n \times [0,\infty). \tag{17}$$

This is well-posed and has a unique solution, provided that the operator $\mathcal{L}$ is semi-bounded (Hesthaven et al., 2007). The solution is given by $u(x,t) = e^{t\mathcal{L}}u_0(x)$. Regarding the 1D heat equation, $\mathcal{L}$ is $c\partial_{xx}$ with diffusivity constant $c$. Then, the solution of the heat equation is given by

$$u(x,t) = \sum_{\xi \in \mathbb{Z}} e^{-4\pi^2 \xi^2 ct} \hat{u}_0(\xi) e^{2\pi i x \xi}. \tag{18}$$

This indicates that the mapping from $u_0(x)$ to $u(x,t)$ is the PDO with the symbol $e^{-4\pi^2 \xi^2 ct}$. Consequently, we propose the time-dependent PDIOs given as follows:

$$\mathcal{K}_a[f](x,t) := a_{\theta_1}^{nn}(x,t)\mathcal{F}^{-1}\left[a_{\theta_2}^{nn}(\xi,t)\mathcal{F}[f](\xi)\right], \tag{19}$$

where $a_{\theta_1}^{nn}(x,t)$ and $a_{\theta_2}^{nn}(\xi,t)$ represent time-dependent symbol networks. For the heat equation, we verify that the time-dependent PDIOs approximate the time-dependent symbol accurately even in a finer time grid than a time grid employed for training. Furthermore, the time-dependent PDIO is applied to obtain the continuous-in-time solution operator of the PDEs (refer to the experiment on the Navier-Stokes equation).

## 5 Experiments

### 5.1 Toy example : 1D heat equation

In this experiment, we verify whether the proposed time-dependent PDIO learns the symbol of the analytic PDO. We consider the 1D heat equation provided in (17). The solution operator of the 1D heat equation is a PDO, which is provided in (18). We attempt to learn the mapping from the initial state and time $(u_0(x),t)$ to the solution $u(x,t)$. The initial state $u_0(x)$ is generated from the Gaussian random field $\mathcal{N}(0, 7^4(-\Delta+7^2)^{-2.5})$ with the periodic boundary conditions. $\Delta$ denotes the Laplacian. The diffusivity constant $c$ and spatial resolution are set to 0.05 and $2^{10} = 1024$, respectively. We utilize 1000 pairs of training data, with inputs at $t = 0$ and outputs at 10 time grids of $t = 0.05 + 0.1n$ ($n = 0, 1, ..., 9$) for each of the 100 initial states. We conduct testing with 20 initial states, using inputs at $t = 0$ and outputs at finer time grids of $t = 0.05 + 0.05n$ ($n = 0, 1, ..., 19$). The time-dependent PDIO (19) is employed and achieves a relative $L^2$ error lower than 0.01 on both the training and test sets. Figure 3 illustrates the symbol network and the analytic symbol provided in (18) on $(\xi,t) \in [-12,12] \times [0.05,1]$. Although the PDIO learns from a sparse time grid, it obtains an accurate symbol for all $t \in [0.05,1]$.

### 5.2 Nonlinear solution operators of PDEs

In this section, we verify the PDNO on a nonlinear PDE dataset. For all the experiments, we employ the PDNO that comprises four layers of the network described in Figure 6 and (16) with a nonlinear activation GELU. Fully connected neural networks are utilized for symbol networks up to layer 3 and hidden dimension 64. The relative $L^2$ error is adopted for the loss function. Detailed hyperparameters are contained in C. We do not truncate the Fourier coefficient in any of the layers, thus indicating that we utilize all of the available frequencies from $-\lceil \frac{s}{2} \rceil$ to $\lceil \frac{s}{2} \rceil - 1$. This is because PDNO does not require additional learning parameters, even if all frequencies are utilized. However, because evaluations are required at numerous grid points, considerable memory is required in the learning process. In practical settings, it is recommended to truncate the frequency space into appropriate maximum modes $k_{max}$. We observed a negligible degradation in performance even when truncation was utilized (refer to F.3). All experiments were conducted using up to five NVIDIA A5000 GPUs with 24 GB memory.

**Benchmark models** We compare the proposed model with the multiwavelet-based model (**MWT**) and the FNO, which are the advanced approaches based on the neural operator architecture. For the difference between **PDNO $(a(x,\xi))$** and **PDNO $(a(\xi))$**, refer to Section 5.3. We conducted the experiments on Darcy flow and the Navier-Stokes equation. Regarding the Navier-Stokes equation, we utilized the same data presented by Li et al. (2020a). For Darcy flow, we regenerated the data according to the same data generation scheme.

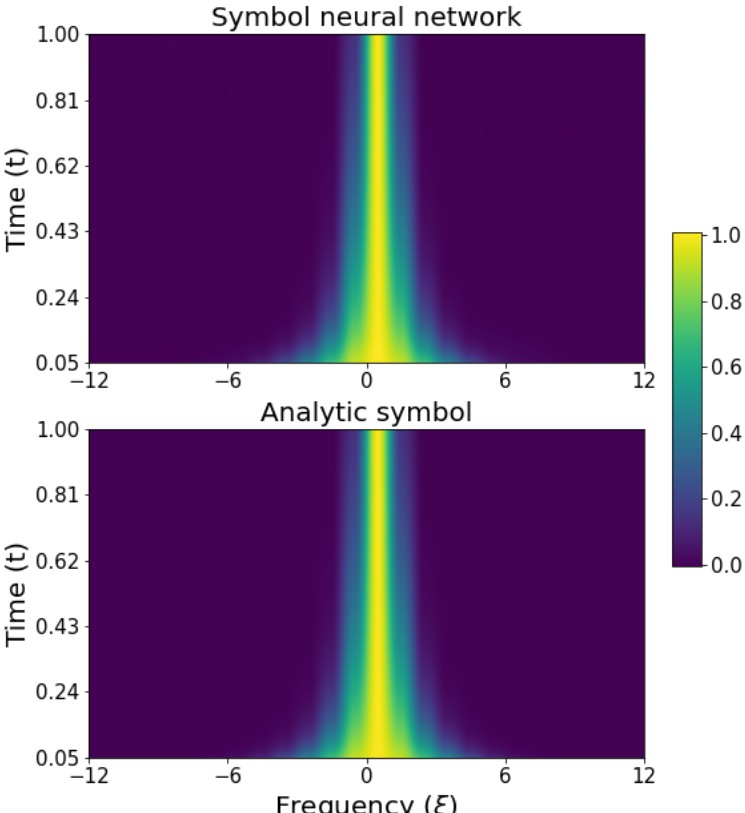

Figure 3: Visualization of the learned symbol from the time-dependent PDIO $a^{nn}_{\theta_1}(x,t)a^{nn}_{\theta_2}(\xi,t)$ (top) and analytic symbol $a(x,\xi,t) = e^{-4\times 0.05\pi^2\xi^2 t}$ (bottom) of the solution operator of the 1D heat equation. Note that learned $a^{nn}_{\theta_1}(x,t)$ is a constant function according to $x$. i.e. $a^{nn}_{\theta_1}(x,t) = c(t)$ (See Figure 7). Therefore, it does not require an $x$-coordinate to plot the learned symbol.

**Darcy flow**  The Darcy flow problem is a diffusion equation with an external force, which describes the flow of a fluid through a porous medium. The steady state of the Darcy flow on the unit box is expressed as

$$\begin{cases} \nabla \cdot (a(x)\nabla u(x)) = f(x), & x \in [0,1]^2 \\ u(x) = 0, & x \in \partial(0,1)^2, \end{cases} \tag{20}$$

where $u$, $a(x)$, and $f(x)$ denote the density of the fluid, diffusion coefficient, and external force, respectively. We attempt to learn the nonlinear mapping from $a(x)$ to the steady state $u(x)$, fixing the external force $f(x) = 1$. The diffusion coefficient $a(x)$ is generated from $\psi_\# \mathcal{N}(0, (-\Delta + 9I)^{-2})$, where $\Delta$ is the Laplacian with zero Neumann boundary conditions, and $\psi_\#$ is the pointwise push forward, defined by $\psi(x) = 12$ if $x > 0$, 3 elsewhere. The coefficient imposes the ellipticity on the differential operator $\nabla \cdot (a(x)\nabla)(\cdot)$. We generate $a(x)$ and $u(x)$ using the second-order FDM on a $512 \times 512$ grid. The lower-resolution dataset is obtained by subsampling. We utilized 1000 training pairs and 100 test pairs and fixed the hyperparameters for all resolutions.

The results on the Darcy flow are presented in Table 1 for various resolutions $s$. The proposed model achieves the lowest relative error for all resolutions. Regarding $s = 32$, particularly, MWT and FNO exhibited the highest errors. Furthermore, the proposed model maintains its performance even at low resolutions.

Table 1: Benchmark (relative $L^2$ error) on Darcy flow on different resolution $s$.

| Resolution | Data | PDNO $(a(x,\xi))$ | PDNO $(a(\xi))$ | MWT Leg | FNO |
|---|---|---|---|---|---|
| $s = 32$ | train | $3.52 \times 10^{-3}$ | $4.08 \times 10^{-3}$ | $1.17 \times 10^{-3}$ | $2.65 \times 10^{-3}$ |
| | test | $\mathbf{3.34 \times 10^{-3}}$ | $3.82 \times 10^{-3}$ | $1.62 \times 10^{-2}$ | $1.78 \times 10^{-2}$ |
| $s = 64$ | train | $2.59 \times 10^{-3}$ | $2.98 \times 10^{-3}$ | $1.81 \times 10^{-3}$ | $2.93 \times 10^{-3}$ |
| | test | $\mathbf{2.52 \times 10^{-3}}$ | $2.85 \times 10^{-3}$ | $1.08 \times 10^{-2}$ | $1.12 \times 10^{-2}$ |
| $s = 128$ | train | $1.58 \times 10^{-3}$ | $2.57 \times 10^{-3}$ | $1.49 \times 10^{-3}$ | $2.77 \times 10^{-3}$ |
| | test | $\mathbf{1.62 \times 10^{-3}}$ | $2.45 \times 10^{-3}$ | $9.27 \times 10^{-3}$ | $1.04 \times 10^{-2}$ |
| $s = 256$ | train | $1.54 \times 10^{-3}$ | $2.62 \times 10^{-3}$ | $1.34 \times 10^{-3}$ | $2.78 \times 10^{-3}$ |
| | test | $\mathbf{1.41 \times 10^{-3}}$ | $2.54 \times 10^{-3}$ | $8.83 \times 10^{-3}$ | $1.01 \times 10^{-2}$ |
| $s = 512$ | train | $1.98 \times 10^{-3}$ | $2.25 \times 10^{-3}$ | $1.32 \times 10^{-3}$ | $2.80 \times 10^{-3}$ |
| | test | $\mathbf{1.93 \times 10^{-3}}$ | $2.17 \times 10^{-3}$ | $9.27 \times 10^{-3}$ | $1.02 \times 10^{-2}$ |

Table 2: Benchmark (relative $L^2$ error) on the Navier-Stokes equation on the various viscosity $\nu$, the time horizon $T$, and the number of data $N$.

| Networks | $\nu = 1e-3$ $T = 50$ $N = 1000$ | $\nu = 1e-4$ $T = 30$ $N = 1000$ | $\nu = 1e-4$ $T = 30$ $N = 10000$ | $\nu = 1e-5$ $T = 20$ $N = 1000$ |
|---|---|---|---|---|
| PDNO $(a(x,\xi))$ | 0.00903 | **0.1320** | 0.0679 | **0.1093** |
| PDNO $(a(x,\xi,t))$ | 0.0299 | 0.2296 | 0.1605 | 0.1852 |
| MWT Leg | **0.00625** | 0.1518 | **0.0667** | 0.1541 |
| MWT Chb | 0.00720 | 0.1574 | 0.0720 | 0.1667 |
| FNO-2D | 0.0128 | 0.1559 | 0.0973 | 0.1556 |
| FNO-3D | 0.0086 | 0.1918 | 0.0820 | 0.1893 |

**Navier-Stokes equation** Navier-Stokes equation describes the dynamics of a viscous fluid. In the vorticity formulation, the incompressible Navier-Stokes equation on the unit torus can be expressed as

$$\begin{cases} \frac{\partial w}{\partial t} + u \cdot \nabla w - \nu \Delta w = f, & (x,t) \in (0,1)^2 \times (0,T], \\ \nabla \cdot u = 0, & (x,t) \in (0,1)^2 \times [0,T], \\ w(x,0) = w_0(x), & x \in (0,1)^2, \end{cases} \tag{21}$$

where $w$, $u$, $\nu$, and $f$ denote the vorticity, velocity field, viscosity, and external force, respectively. We utilize the same Navier-Stokes data used in Li et al. (2020a) to learn the nonlinear mapping from $w(x,0), ...., w(x,9)$ to $w(x,10), ..., w(x,T)$, fixing the force $f(x) = 0.1(\sin(2\pi(x_1+x_2)) + \cos(2\pi(x_1+x_2)))$. The initial condition $w_0(x)$ is sampled from $\mathcal{N}(0, 7^{1.5}(-\Delta + 7^2 I)^{-2.5})$ with periodic boundary conditions. We experiment with four Navier-Stokes datasets:$(\nu, T, N) = (10^{-3}, 50, 1000), (10^{-4}, 30, 1000), (10^{-4}, 30, 10000)$, and $(10^{-5}, 20, 1000)$, where $\nu$, $T$, and $N$ denote the viscosity, final time to predict, and number of training samples, respectively. Notably, the lower the viscosity, the more difficult the prediction. All datasets comprise $64 \times 64$ resolutions.

We employ a recurrent architecture to propagate along the time domain. From $w(x,0), ...., w(x,9)$, the model predicts the vorticity at $t = 10$, $\bar{w}(x,10)$. Then, from $w(x,1), ..., w(x,9), \bar{w}(x,10)$, the model predicts the next vorticity $\bar{w}(x,11)$. We repeat this process until $t = T$.

For each experiment, we utilize 200 test samples. For $(\nu, T, N) = (10^{-3}, 50, 1000)$, we utilize a batch size of 10 or 20 otherwise. Furthermore, we empoly fixed hyperparameters for the four experiments.

The results of the Navier-Stokes equation are presented in Table 2. In all four datasets, the proposed model exhibits comparable or superior performances. Notably, the relative error improves considerably for

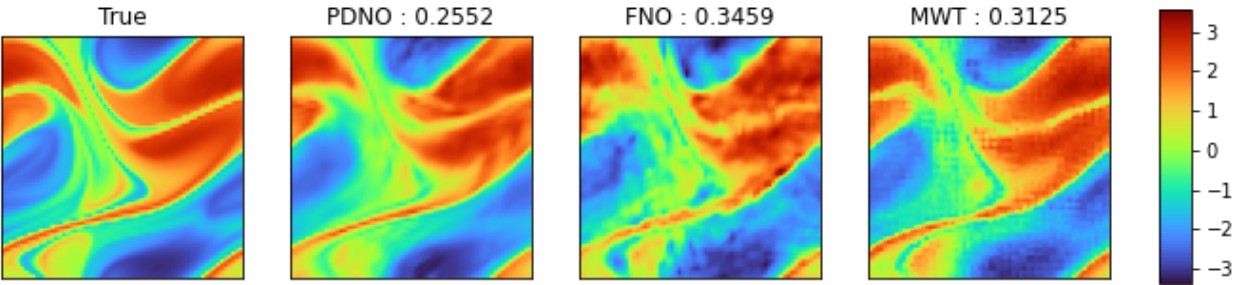

Figure 4: Example of a prediction on the Navier-Stokes data with $\nu$=1e-5 showing the prediction $w(x, 19)$ from inputs $[w(x, 0), ..., w(x, 9)]$. Each value on the top of the figure is the relative $L^2$ error between the true $w(x, 19)$ and each prediction. A prediction of FNO is more granular than PDNO. We suspect that this is due to non-smooth symbols of FNO.

Table 3: Benchmark (relative $L^2$ error) on Burgers' equation on different resolution $s$.

| Equation | Resolution | PDNO ($a(x,\xi)$) | PDNO ($a(\xi)$) | MWT Leg | MWT Chb | FNO |
|---|---|---|---|---|---|---|
| | $s = 256$ | $6.85 \times 10^{-4}$ | $9.03 \times 10^{-4}$ | $1.99 \times 10^{-3}$ | $4.02 \times 10^{-3}$ | $\mathbf{6.49 \times 10^{-4}}$ |
| | $s = 512$ | $8.49 \times 10^{-4}$ | $1.22 \times 10^{-3}$ | $1.85 \times 10^{-3}$ | $3.81 \times 10^{-3}$ | $\mathbf{6.47 \times 10^{-4}}$ |
| Burgers | $s = 1024$ | $1.10 \times 10^{-3}$ | $1.25 \times 10^{-3}$ | $1.84 \times 10^{-3}$ | $3.36 \times 10^{-3}$ | $\mathbf{6.40 \times 10^{-4}}$ |
| | $s = 2048$ | $1.18 \times 10^{-3}$ | $1.29 \times 10^{-3}$ | $1.86 \times 10^{-3}$ | $3.95 \times 10^{-3}$ | $\mathbf{6.47 \times 10^{-4}}$ |
| | $s = 4096$ | $1.78 \times 10^{-3}$ | $1.91 \times 10^{-3}$ | $1.85 \times 10^{-3}$ | $2.99 \times 10^{-3}$ | $\mathbf{6.53 \times 10^{-4}}$ |

$(\nu, T, N) = (10^{-5}, 20, 1000)$, thereby exhibiting the lowest viscosity. Figure 4 displays a sample prediction at $t = 19$, which is highly unpredictable.

### 5.3 Additional experiments

**Darcy flow.** On Darcy flow, we perform an additional experiment, which does not utilize a symbol network $a_{\theta_1}^{nn}(x)$, but $a_{\theta_2}^{nn}(\xi)$. In this case, the PDNO has the same structure as FNO except for symbol networks. Refer to PDNO ($a(\xi)$) in Table 1. Although less than the original PDNO, the results of the PDNO without the dependency of the $x$-symbol perform better than the other models, including FNO. This explains why the smoothness of the symbol of PDNO is important.

**Navier-Stokes equation.** On the Navier-Stokes equation, we also present the results with time-dependent PDIO (Section 4.2) in Table 2. This exhibits a relatively high error but has the advantage of not using a recursive structure. In addition, for the Navier-Stokes equation with $\nu = 1e-5$, we compare the training and test relative $L^2$ errors along time $t$ in Figure 1. All models demonstrate that the test errors grow exponentially according to time $t$. Among them, PDNO consistently demonstrates the least test errors for all time $t$. More notable is the difference between the solid and dashed lines, which indicates that MWT and FNO suffer from overfitting, whereas PDNO does not. The same trend is observed for Darcy flow (refer Table 1). This might be related to the smoothness of the models' symbols. Furthermore, the symbols of PDNO and FNO are visualized in Figure 5.

**Non-smooth solution operator for Burgers' equation.** We expect that the PDNO based on the PDO theory will provide a more accurate operator approximation compared to the conventional FNO when handling a smooth solution operator. To obtain a more precise understanding, we conducted additional experiments to approximate the solution operator of the Burgers' equation. The 1D Burgers' equation is a nonlinear PDE, which describes the interaction between the effects of nonlinear convection and diffusion as

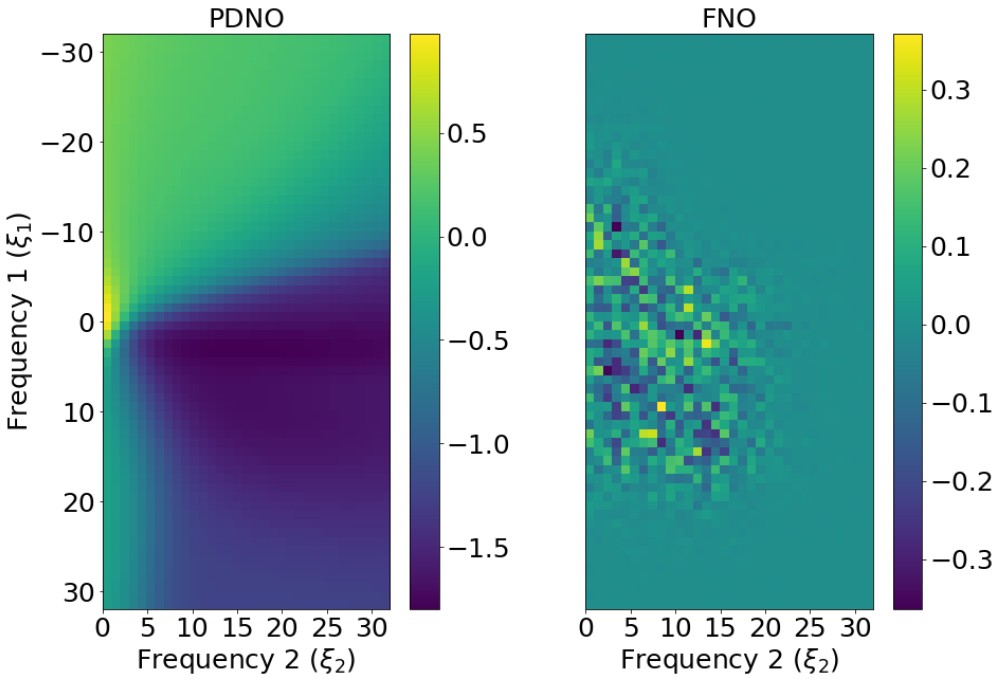

Figure 5: Examples of the real part of learned symbol $a_{ij}^{nn}(\xi)$ from the Navier-Stokes data with $\nu = 1e - 5$. $x$-axis and $y$-axis represent frequency domains. As we used real valued functions, the second coordinate is half the first.

follows:

$$\begin{cases} \frac{\partial u}{\partial t} = -u \cdot \frac{\partial u}{\partial x} + \nu \frac{\partial^2 u}{\partial x^2}, & (x, t) \in (0, 1) \times (0, 1], \\ u(x, 0) = u_0(x), & x \in (0, 1), \end{cases} \tag{22}$$

where $u_0$ is the initial state and $\nu$ denotes the viscosity. We attempt to learn the nonlinear mapping from the initial state $u_0(x)$ to the solution $u(x, 1)$ at $t = 1$. The Burgers' equation with a small viscosity parameter $\nu$ generates sharp shocks over time. In such cases, the solution operator becomes less smooth, and it is conceivable that in such scenarios, PDNO may provide a less accurate operator approximation than the FNO. The results from the Burgers data are presented in Table 3 along with different resolutions $s$. As expected, in the case of the Burgers' equation with non-smooth solutions, the PDNO demonstrated errors that were either comparable to or even higher than those of the FNO. In addition to theoretical analysis in Section 3, this additional experiment confirms that PDNO is more beneficial when approximating smoother and more continuous solution operators. Refer to Appendix F.2 for more details.

## 6   Conclusion

Based on the PDO theory, we developed a novel PDIO and PDNO framework that efficiently learns mappings between function spaces. The proposed symbol networks are in a toroidal symbol class that renders the corresponding PDIOs continuous between Sobolev spaces on the torus, which can considerably improve the learning of the solution operators in most experiments. This study revealed an excellent ability for learning operators based on the theory of PDO. However, there is room for improvement in highly complex PDEs such as the Navier-Stokes equation, and the time-dependent PDIOs are difficult to apply to nonlinear architecture. We expect to solve these problems by employing advanced operator theories (Duistermaat, 1996; Hörmander, 1971; Duistermaat & Hörmander, 1972), which will ultimately address engineering and physical problems.

**Acknowledgments**

Jin Young Shin and Hyung Ju Hwang were supported by the National Research Foundation of Korea(NRF) grant funded by the Korea government(MSIT) (No. RS-2023-00219980 and RS-2022-00165268) and by Institute for Information & Communications Technology Promotion (IITP) grant funded by the Korea government(MSIP) (No.2019-0-01906, Artificial Intelligence Graduate School Program (POSTECH)). Jae Yong Lee was supported by a KIAS Individual Grant (AP086901) via the Center for AI and Natural Sciences at Korea Institute for Advanced Study and by the Center for Advanced Computation at Korea Institute for Advanced Study.

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

## A    Notations

The main notations employed throughout this paper are presented in Table 4.

Table 4: Notations

| Notations | Descriptions |
|---|---|
| $\mathcal{A}$ | an input function space |
| $\mathcal{U}$ | an output function space |
| $\mathcal{G} : \mathcal{A} \to \mathcal{U}$ | an operator from $\mathcal{A}$ to $\mathcal{U}$ |
| $x \in \mathbb{R}^n$ or $\mathbb{T}^n$ | a variable in the spatial domain |
| $\xi \in \mathbb{R}^n$ or $\mathbb{Z}^n$ | a variable in the Fourier space |
| $\hat{f}(\xi)$ | Fourier transform of function $f(x)$ |
| $S_{\rho,\delta}^m$ | an Euclidean (or toroidal) symbol class |
| $a(x,\xi) \in S_{\rho,\delta}^m$ | an Euclidean (or toroidal) symbol |
| $a_\theta^{nn}(x,\xi)$ | a symbol network parameterized by $\theta$ |
| $T_a : \mathcal{A} \to \mathcal{U}$ | a PDO with the symbol $a(x,\xi)$ |
| $\mathcal{K}_a : \mathcal{A} \to \mathcal{U}$ | a PDIO with a symbol network $a_\theta(x,\xi)$ |
| $\mathcal{K}_R : \mathcal{A} \to \mathcal{U}$ | a Fourier integral operator (Li et al., 2020a) |
| $\mathcal{F} : \mathcal{A} \to \mathcal{U}, \mathcal{F}^{-1} : \mathcal{U} \to \mathcal{A}$ | Fourier transform and its inverse |
| $\|\xi\|$ | Euclidean norm |
| $\langle \xi \rangle$ | $(1 + \|\xi\|^2)^{\frac{1}{2}}$ |
| $\triangle_\xi^\alpha$ | a difference operator of order $\alpha$ on $\xi$ |
| $k_{max}$ | the maximum number of Fourier modes |

## B    Analysis on symbol $a(x,\xi)$

### B.1    Decomposable assumption $a^{nn}(x,\xi) = a^{nn}(x)a^{nn}(\xi)$

The symbol was assumed to be decomposable because of computational costs. With a decomposable symbol network $a^{nn}(x)a^{nn}(\xi)$, we only needed one IFFT computation

$$\mathcal{K}_a(f)(x) = a^{nn}(x) \underbrace{\sum_{\xi \in \mathbb{Z}^n} a^{nn}(\xi)\hat{f}(\xi)e^{2\pi i \xi x}}_{\text{IFFT}} = a^{nn}(x)\mathcal{F}^{-1}\left[a^{nn}(\xi)\hat{f}(\xi)\right].$$

With the symbol network $a^{nn}(x,\xi)$, an input of IFFT depends on the spatial domain $x$. Hence, we need IFFT computations as many times as the number of grids in the spatial domain $x$.

$$\mathcal{K}_a(f)(x) = \underbrace{\sum_{\xi \in \mathbb{Z}^n} a^{nn}(x,\xi)\hat{f}(\xi)e^{2\pi i \xi x}}_{\text{IFFT}} = \mathcal{F}^{-1}\left[a^{nn}(x,\xi)\hat{f}(\xi)\right].$$

Therefore, the non-decomposable symbols require the number of $x$ grid points multiplied by the cost of decomposable symbols.

### B.2    Reason for considering the symbol network on domain $\mathbb{T}^n \times \mathbb{Z}^n$

In this section, we explain why the proposed model should be addressed in $\mathbb{T}^n \times \mathbb{Z}^n$ instead of $\mathbb{T}^n \times \mathbb{R}^n$. For convenience, we assume that $n = 1$. Let $f : \mathbb{T} \to \mathbb{R}$ and its $N$ points discretization $f(\frac{1}{N}) = y_0$, $f(\frac{2}{N}) = y_1$, ... $f(\frac{N}{N}) = y(1) = y_{N-1}$. Then, the DFT of the sequence $\{y_n\}_{0 \le n \le N-1}$ is expressed as

$$\xi_k = \frac{1}{N}\sum_{n=0}^{N-1} y_n e^{-2\pi i k \frac{n}{N}} \tag{23}$$

and the inverse DFT of $\{\xi_k\}_{0 \leq k \leq N-1}$ is expressed as

$$y_n = \sum_{k=0}^{N-1} \xi_k e^{2\pi i k \frac{n}{N}}. \tag{24}$$

As $N$ tends to $\infty$, it can be deduced that

$$\lim_{N \to \infty} \xi_k = \lim_{N \to \infty} \frac{1}{N} \sum_{n=0}^{N-1} f(\frac{n+1}{N}) e^{-2\pi i k \frac{n}{N}} \to \int_0^1 f(x) e^{-2\pi i k x} \, dx = \hat{f}(k)$$

and

$$\lim_{N \to \infty} y_n = \lim_{N \to \infty} f(\underbrace{\frac{n}{N}}_{x} + \frac{1}{N}) = \lim_{N \to \infty} \sum_{k=0}^{N-1} \xi_k e^{2\pi i k \frac{n}{N}} \to \sum_{k=0}^{\infty} \hat{f}(k) e^{2\pi i k x} = f(x),$$

where $x = \frac{n}{N}$. Hence, DFT is an approximation of integral on $\mathbb{T}$ and IDFT is an approximation of the infinity sum on $\mathbb{Z}$. Therefore, the PDO theory on $\mathbb{T}^n \times \mathbb{Z}^n$ is more suitable for our model.

## C   Hyperparameters

Table 5: Hyperparameters for learning PDNOs on each dataset. # layers, # hidden and activation are for symbol networks.

| Data | Batch size | LR | Weight decay | Epochs | Step size | # Channel | # Layers | # Hidden | Activation |
|------|-----------|-----|-------------|--------|-----------|-----------|----------|----------|-----------|
| Heat | 20 | $1e^{-2}$ | $1e^{-6}$ | 10000 | 2000 | 1 | 2 | 40 | GELU |
| Darcy | 20 | $1e^{-2}$ | $1e^{-6}$ | 1000 | 200 | 20 | 3 | 32 | GELU |
| N-S | 20 | $5e^{-3}$ | $1e^{-6}$ | 1000 | 200 | 20 | 2 | 32 | GELU |
| Burgers | 20 | $5e^{-3}$ | 0 | 1000 | 100 | 64 | 2 | 64 | TANH |

## D   Resource requirements

Table 6: Resource comparison of PDNO and FNO on NS data. PDNO uses a two-layer symbol network with hidden dimension 32. The memory requirements are obtained from *nvidia-smi* command. We used a single NVIDIA A5000 GPU.

| Model | $k_{max}$ | # Channel | Memory (train) | # Parameter | Time (sec/epoch) |
|-------|-----------|-----------|----------------|-------------|------------------|
| PDNO | 12 | 20 | 9939 MB | $1.90 \times 10^5$ | 20.77 |
| PDNO | 12 | 30 | 17819 MB | $3.91 \times 10^5$ | 40.73 |
| PDNO | 32 | 20 | 10143 MB | $1.90 \times 10^5$ | 22.05 |
| FNO | 12 | 20 | 3625 MB | $4.66 \times 10^5$ | 4.58 |
| FNO | 12 | 30 | 3941 MB | $1.05 \times 10^6$ | 5.49 |
| FNO | 32 | 20 | 3957 MB | $3.28 \times 10^6$ | 4.80 |

In Table 6, we compared the memory requirement, the number of parameters, and the training time of PDNO and FNO. PDNO requires more memory than FNO in training because it needs to compute the symbol networks $a^{nn}(x)$ and $a^{nn}(\xi)$. However, PDNO has fewer parameters than FNO; hence, it requires lower storage to save the trained model. If a faster inference is required, the evaluated values of symbol network may be stored. To reduce memory resources and time consumption during training, one possible future work is to smoothen the symbol network via regularization on the parametric symbol $R_{ij}$ in 3.

# E    Activation functions for symbol network

In this section, we discuss the activation function for the symbol network. We proved the Proposition 1 when the GELU activation function was utilized for the symbol network. In addition to GELU, other activation functions can be used for the symbol network. To explain this, we first define the Schwartz space (Reed & Simon, 1972) as follows:

**Definition 5.** *The Schwartz space $\mathcal{S}(\mathbb{R}^n)$ is the topological vector space of functions $f : \mathbb{R}^n \to \mathbb{C}$ such that $f \in C^\infty(\mathbb{R}^n)$ and*

$$z^\alpha \partial^\beta f(z) \to 0, \quad as \ |z| \to \infty, \tag{25}$$

*for every pair of multi-indices $\alpha, \beta \in \mathbb{N}_0^n$.*

In other words, the Schwartz space comprises smooth functions whose derivatives decay at infinity faster than any power. As mentioned in the proof of Proposition 1, it can be easily demonstrated that the second or higher derivatives of GELU are in the Schwartz space $\mathcal{S}(\mathbb{R})$. Because GELU is defined as $\sigma(z) = z\Phi(z)$ with $\Phi(z) = \frac{1}{\sqrt{2\pi}} \int_{-\infty}^{z} \exp(-u^2/2)du$, the second or higher derivatives of GELU is the sum of exponential decay functions $\exp(-z^2/2)$. Hence, the second or higher derivatives of the function are in the Schwartz space, i.e., $\sigma^{(k)} \in \mathcal{S}(\mathbb{R})$ when $k \geq 2$.

Next, we prove that another activation function $\phi(z)$ is in symbol class $S_{1,0}^1(\mathbb{T}^n \times \mathbb{R}^n)$ if the difference between the function $\phi(z)$ and GELU $\sigma(z)$ is in the Schwartz space. We refer to a function like $\phi(z)$ as the GELU-like activation function. It can be easily demonstrated that the function $\phi(z)$ is bounded by the linear function $|z|$ because GELU is bounded by the linear function. Because the Schwartz space is closed under differentiation, $\phi(z) - \sigma(z) \in \mathcal{S}(\mathbb{R})$ implies $\phi^{(k)}(z) - \sigma^{(k)}(z) \in \mathcal{S}(\mathbb{R})$ for $k \in \mathbb{N}$. Because GELU satisfies $\sigma'(z) \leq c_\alpha$ and $\sigma^{(k)} \in \mathcal{S}(\mathbb{R})$ when $k \geq 2$, the activation function $\phi(z)$ also satisfies $\phi'(z) \leq c_\alpha$ and $\phi^{(k)} \in \mathcal{S}(\mathbb{R})$ when $k \geq 2$. Therefore, the proof of Proposition 1 can be obtained by altering another activation function $\phi(z)$ instead of GELU $\sigma(z)$. GELU-like activation functions, such as the Softplus (Glorot et al., 2011), and Swish (Ramachandran et al., 2017) etc., satisfy the aforementioned assumption, such that it can be used for the symbol network in our PDIO.

We can easily demonstrate that the symbol network $a_\theta^{nn}(x,\xi)$ with $\tanh(z) = \frac{e^z - e^{-z}}{e^z + e^{-z}}$ is in $S_{1,0}^0(\mathbb{T}^n \times \mathbb{R}^n)$. In the proof of Proposition 1, we adopted the characteristic of GELU and its high derivatives. The tanh function is bounded and the first or higher derivatives of the tanh function are in the Schwartz space. Therefore, neural network $a_{\theta_2}^{nn}(\xi)$ satisfies the following boundedness:

$$|\partial_\xi^\alpha a_{\theta_2}^{nn}(\xi)| \leq c_\alpha, \quad \text{if } |\alpha| = 0, \tag{26}$$

$$|\partial_\xi^\alpha a_{\theta_2}^{nn}(\xi)| \leq c_\alpha \langle\xi\rangle^{-|\alpha|}, \quad \text{if } |\alpha| \geq 1. \tag{27}$$

Note that the boundedness of the neural network $a_{\theta_1}^{nn}(x)$ is same in the case of GELU. Hence, we can bound the derivative of the symbol network $a_\theta^{nn}(x,\xi)$ as

$$|\partial_x^\beta \partial_\xi^\alpha a(x,\xi)| \leq c_{\alpha\beta} \langle\xi\rangle^{-|\alpha|}. \tag{28}$$

Therefore, the symbol network with the tanh activation function is in $S_{1,0}^0(\mathbb{T}^n \times \mathbb{R}^n)$. Similarly, it is easy to prove that the sigmoid function $\frac{1}{1+e^{-z}}$ is also in a symbol class $S_{1,0}^0(\mathbb{T}^n \times \mathbb{R}^n)$. Therefore, the PDIOs with these two activation functions are bounded linear operators from the Sobolev space $W^{p,s}(\mathbb{T}^n)$ to the Sobolev space $W^{p,s}(\mathbb{T}^n)$ for all $1 < p < \infty$ and any $s \in \mathbb{R}$.

# F    Additional figures and experiments

## F.1    1D heat equation : Symbol network $a_{\theta_1}^{nn}(x,t)$.

In 1D heat equation experiments, we assume that the symbol is decomposed by $a(x,\xi,t) \approx a_{\theta_1}^{nn}(x,t) \times a_{\theta_2}^{nn}(\xi,t)$. Figure 7 illustrates the learned symbol network $a_{\theta_1}^{nn}(x,t)$ on $(x,t) \in \mathbb{T} \times [0.05, 1]$.Evidently, $a_{\theta_1}^{nn}(\cdot,t)$ is almost

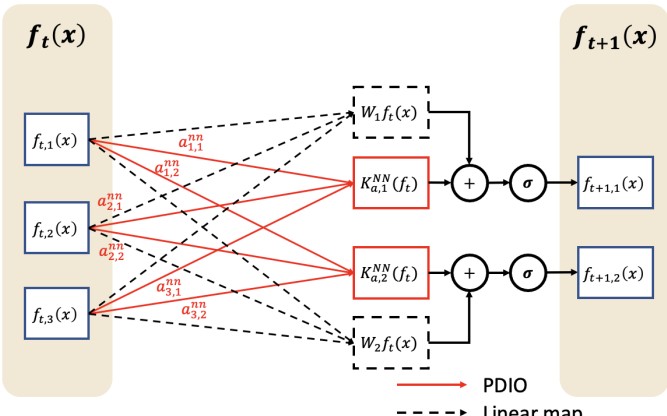

Figure 6: Structure of (1) using the integral operator $\mathcal{K}_a$ in (16) with $c_{in} = 3$ and $c_{out} = 2$. Each black solid line represents a PDIO with symbol network $a_{ij}^{nn}$.

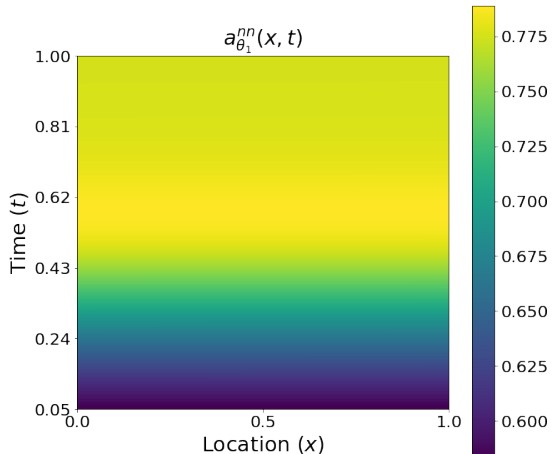

Figure 7:   Learned symbol $a_{\theta_1}^{nn}(x, t)$ from 1D heat equation.

a constant function for each $t \in [0.05, 1]$. Accordingly, $a_{\theta_1}^{nn}(x, t)$ is considered a function of $t$ by taking the average along the $x$-dimension to visualize $a_{\theta_1}^{nn}(x, t) a_{\theta_2}^{nn}(\xi, t)$ in Figure 3.

In addition, Figure 8 visualizes a sample prediction on the 1D heat equation.

### F.2   Details of experiments on 1D Burgers' equation.

Here, we employ the same Burgers' data utilized in Li et al. (2020a). The initial state $u_0(x)$ is generated from the Gaussian random field $\mathcal{N}(0, 5^4(-\Delta + 25I)^{-2})$ with the periodic boundary conditions. The viscosity $\nu$ and finest spatial resolution are set to 0.1 and $2^{13} = 8192$, respectively. The lower-resolution dataset is obtained via subsampling. We experiment with the same hyperparameters for all resolutions. In addition, we utilize 1000 train pairs and 100 test pairs.

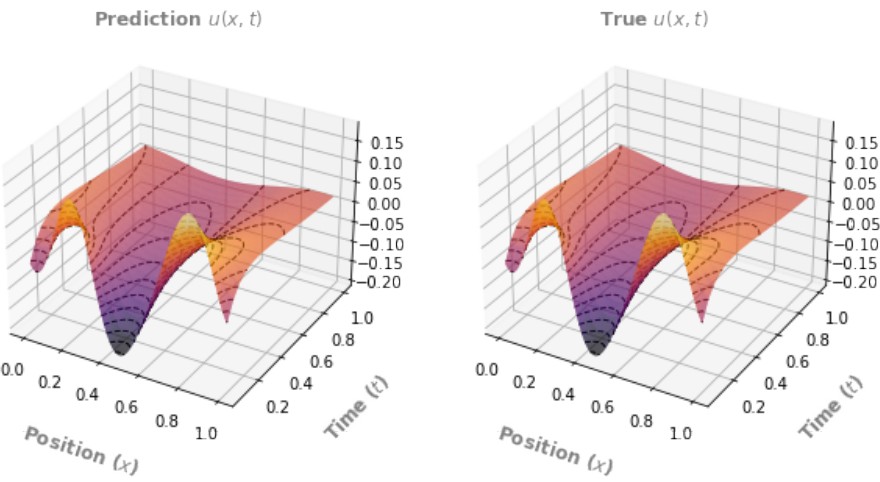

Figure 8: A sample of prediction on 1D heat equation from a PDIO. The model is trained on $1024 \times 10$ dataset and evaluated on $1024 \times 20$. Dashed lines on the surface are contour lines.

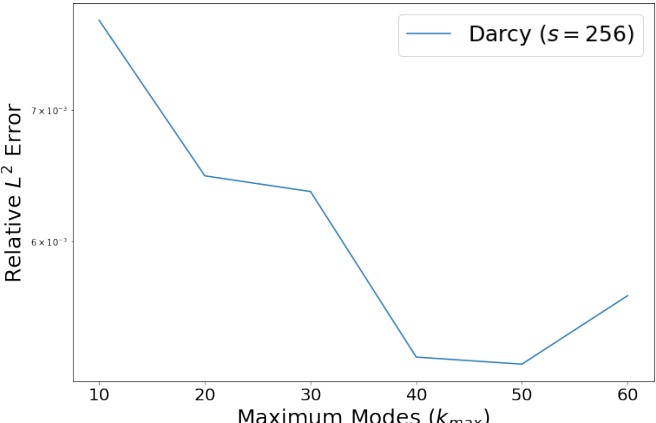

Figure 9: Test relative $L^2$ error that depends on maximum modes $k_{max}$ of PDNO on Darcy flow (resolution $s = 256$).

### F.3 Changes in errors according to $k_{max}$.

As mentioned in Section 5.2, we utilize all possible modes. Although PDNO does not require additional parameters to employ all modes, it demands more memory in the learning process. Hence, we conduct additional experiments on Darcy flow by limiting the number of modes $k_{max}$. In Figure 9, changes in test relative $L^2$ error along $k_{max}$ are illustrated. Even with small $k_{max}$, it still outperforms MWT and FNO (Table 1). In addition, for $k_{max} \geq 20$, PDNO obtains a comparably relative $L^2$ error on the Darcy flow problem.

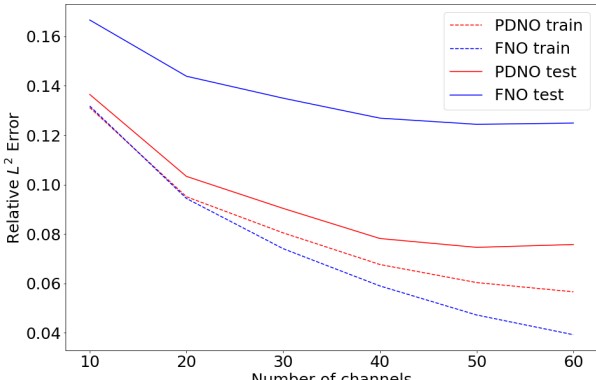

Figure 10: Train and test error of the proposed model and FNO on Navier-Stokes data with $\nu = 1e - 5$ according to the number of channels.

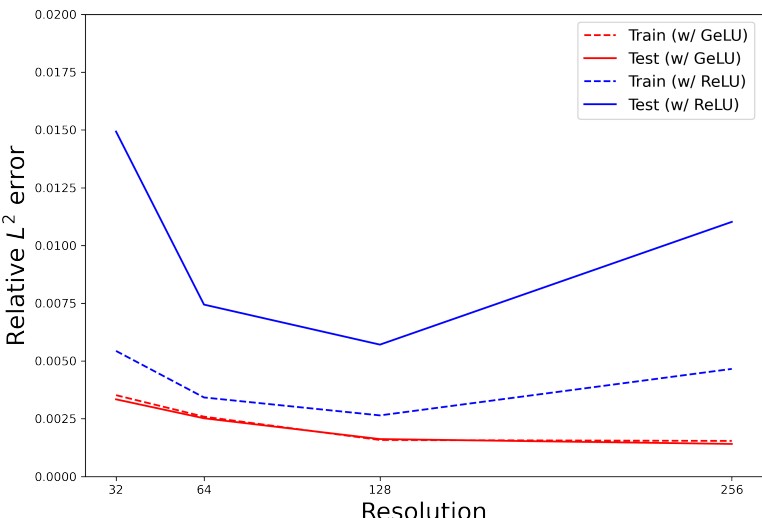

Figure 11: Training and test errors of PDNO on Darcy flow according to the resolution of data using GELU and ReLU activation functions for the symbol network.

### F.4 Changes in activation of the symbol network.

We attempt to investigate how experimental results differ when using a Rectified Linear Unit (ReLU) activation function, as opposed to the smooth activation functions discussed in Appendix E, for the symbol network. Figure 11 illustrates the training and test relative $L^2$ errors as data resolution varies when employing GELU and ReLU activation functions in the symbol network for the Darcy problem. As demonstrated in Proposition 1, using the GELU activation function helps reduce the overfitting of the solution operator, while employing the ReLU activation function results in a notable disparity between the training and test relative $L^2$ errors.

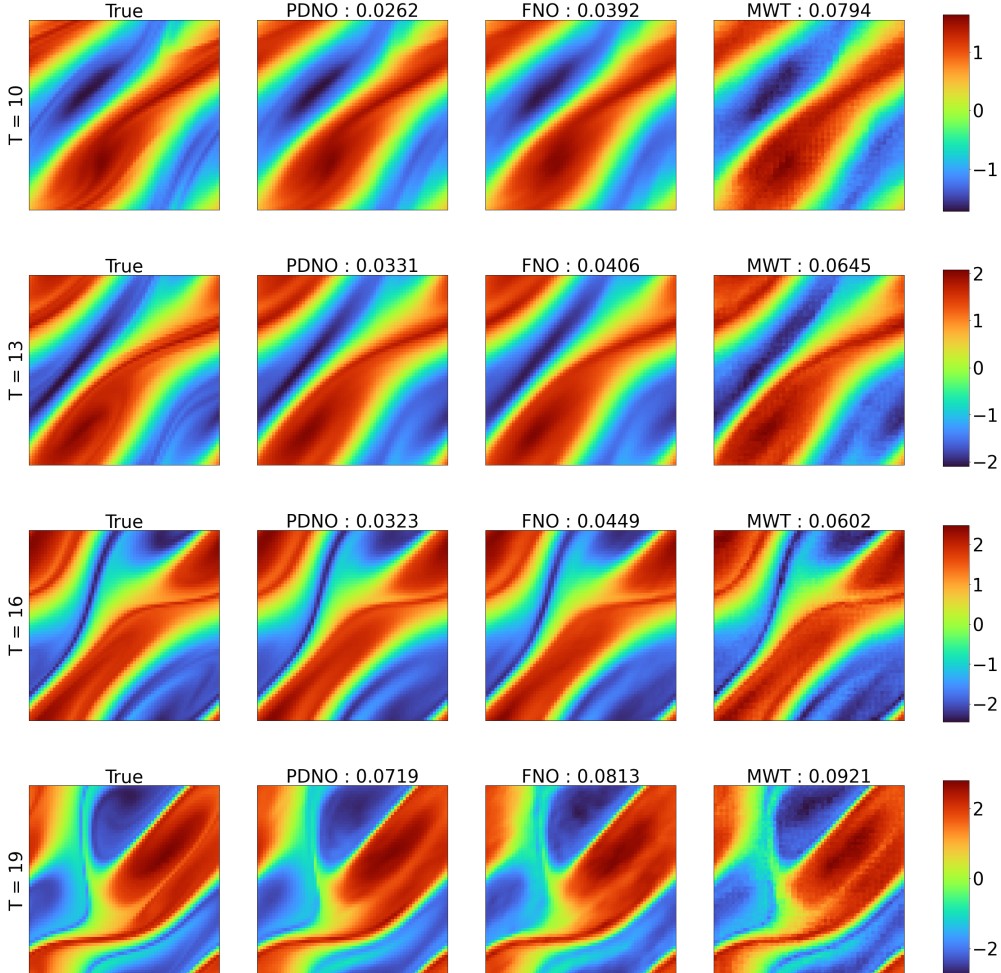

Figure 12: Comparison of the prediction on Navier-Stokes equation with $\nu = 1e - 5$. This test sample shows the lowest relative $L^2$ error on average of three models.

### F.5    Navier-Stokes equation with $\nu = 1e - 5$

**Samples with the lowest and highest error**    Figures 12 and Figures 13 present the samples with the highest and the lowest errors, respectively. PDNOs consistently obtain the lowest error at all time steps of both samples.

**PDNO and FNO with different number of channels**    We compare the performance of PDNO and FNO, which varies depending on the number of channels. For a fair comparison, the truncation is not utilized in the Fourier space for both FNO and PDNO. Furthermore, PDNO utilizes only a single symbol network $a_{\theta_2}^{nn}(\xi)$, not $a_{\theta_1}^{nn}(x)$. In Figure 10, as the number of channels increases, the test error decreases in both models. PDNO achieves lower test errors than FNO and also exhibits a negligible gap between the training and test errors.

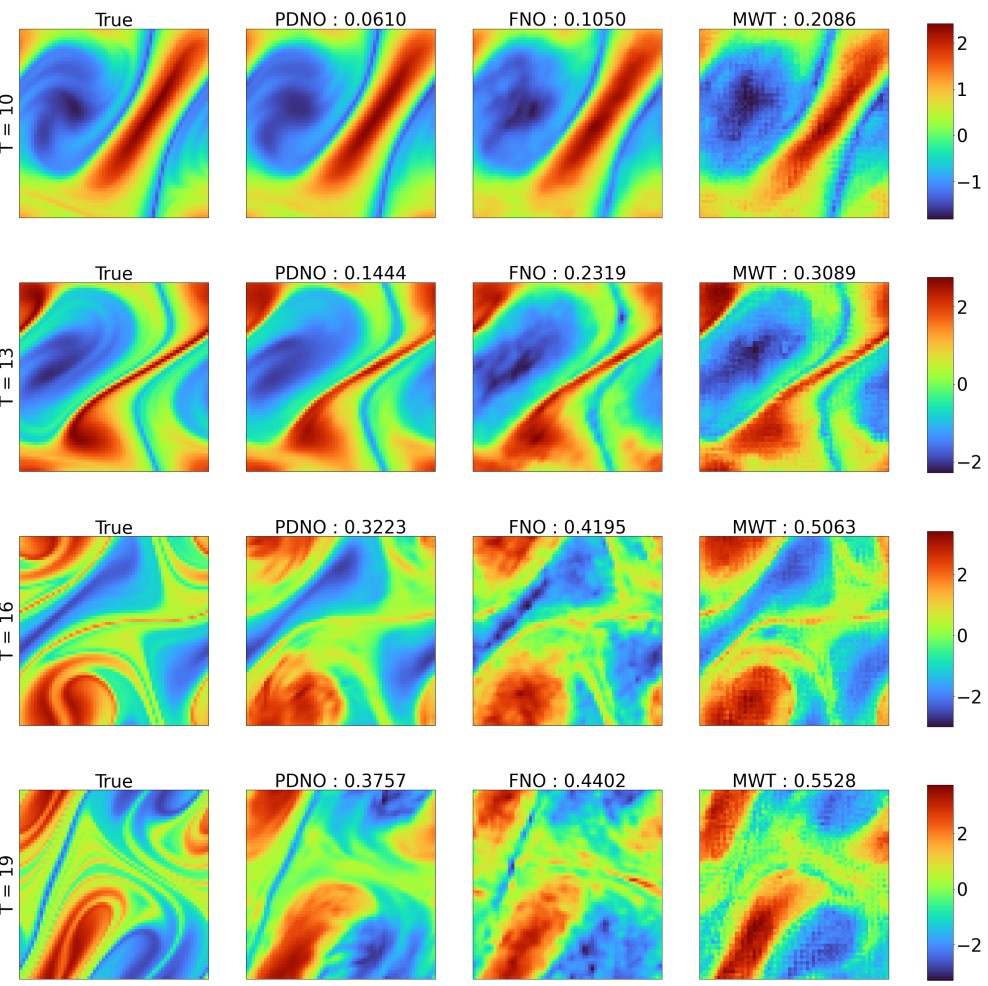

Figure 13: Comparison of the prediction on Navier-Stokes equation with $\nu = 1e - 5$. This test sample shows the greatest relative $L^2$ error on average of three models.

