# OpenReview forum: "Pseudo-Differential Neural Operator: Generalize Fourier Neural operator for Learning Solution Operators of Partial Differential Equations"
_TMLR — Accepted by TMLR_

### Review · Reviewer_fuUu · 2023-06-27

**Summary Of Contributions:**

Pseudo-differential method for operator learning


This paper studies the problem of operator learning between function spaces. Inspired by pseudo-differential operation, the paper proposes a variant of Fourier neural operators that applies a pointwise multiplication after the inverse Fourier transform.

The paper provides a solid inspiration for the proposed method and backs the study with relevant empirical studies.



**Audience:**

Yes

**Broader Impact Concerns:**

The work has broader impact to the operator learning studies.

**Claims And Evidence:**

Yes

**Requested Changes:**

Acknowledge the prior work considered the MLP setting. Moreover, a comparison with MLP based model is helpful.

**Strengths And Weaknesses:**


Strengths:
- The approach in this paper is theoretically and historically motivated.
- The empirical results are appreciated.

Weaknesses:
- The proposed method suggests multiplying the output of the inverse Fourier transform in FNO layers with a function. Much more complex pointwise operations are already considered in prior works and prior codebases. For example,
Adaptive Fourier Neural Operators: Efficient Token Mixers for Transformers
and
Fourcastnet: A global data-driven high-resolution weather model using adaptive Fourier neural operators
consider putting the pointwise MLP layer after the inverse Fourier transfer layer, which is regarded as more complex than the proposed multiplication.
Furthermore, such MLP options are heavily adopted in neural operator codebases, I guess as a default mode.
https://github.com/neuraloperator/neuraloperator/blob/main/neuralop/models/fno_block.py#L10

---

### Review · Reviewer_LJcT · 2023-07-03

**Summary Of Contributions:**

The authors propose a generalization of an integral kernel of the Fourier Neural Operator (FNO) called Pseudo-Differential Neural Operator (PDNO).

The original FNO kernel has a form
$$
\text{FNO}[f^{\beta}] = \text{FFT}^{-1}\left[\sum\_{\beta}A_{j}^{\alpha\beta}\left(\text{FFT}\left[f^{\beta}(x\_{i})\right] (\xi\_{j})\_{j=1:k\_\max}\right) \right] (x\_i),
$$
$$
f^{\beta}(x_i) \in \mathbb{R}^{N_\text{features in}\times N_x}, ~ A_j^{\alpha\beta}\in \mathbb{C}^{k_\max\times N_\text{features out}\times N_\text{features in}}.
$$

In the present article, the authors used

$$
\text{PDNO}[f^{\beta}] =\sum_{\beta} b_{i}^{\alpha\beta}\left(\text{FFT}^{-1}\left[a_{j}^{\alpha\beta}\left(\text{FFT}\left[f^{\beta}(x_{i})\right] (\xi_{j})\right) \right] (x_i)\right),
$$
$$
f^{\beta}(x_i) \in \mathbb{R}^{N_\text{features in}\times N_x}; ~ b_i^{\alpha\beta}, a_j^{\alpha\beta}\in \mathbb{C}^{N_x\times N_\text{features out}\times N_\text{features in}},
$$
where both $a$ and $b$ are generated by feedforward neural networks evaluated at the grid points in frequency and coordinate domains respectively.

In words, in comparison to FNO, there are several novelties:
1. No truncation in frequency space (in some experiments truncation is present).
2. Weights for linear layers are generated by neural networks with a separate set of trainable parameters.
3. There is an additional linear transformation that explicitly depends on $x$.

The authors provide several theoretical results on the relations between the properties of neural networks used to generate weights and the smoothness of the resulting neural operator.

According to the author's experiments
1. PDNO performs better than the multiwavelet neural operator and origin FNO on elliptic partial differential equation with Dirichlet boundary conditions
2. ditto in most settings for the Navier-Stokes equation on the torus
3. on the same two problems proposed operator has a better generalization gap

**Audience:**

Yes

**Claims And Evidence:**

Yes

**Requested Changes:**

I consider all the issues mentioned in this section to be critical. In my opinion current paper warrants publication in TMLR after a suitable revision.

I encourage authors to address the questions below, clarify/correct highlighted pieces, or answer with a rebuttal.

**Motivation**
1. What is the problem? Why do we need to consider new architecture? Reasons that I found scattered in the article:

    i. FNO overfits. Why? What is the evidence for that?

    ii. FNO is not general enough to represent simple time-dependent operators. Is that true? It does not seem to be the case when FNO is used to infer spatiotemporal physical fields, i.e., it applies to the function $u(x, t)$.

    iii. FNO does not constitute a smooth operator in the Banach space. Why is this important? Should we have smooth operators?

    iiii. Why PDNO helps? How and why is it going to solve the problems above?

2. Why do we need the theoretical statements about the symbol of the operator? Suppose we have a continuous operator between Sobolev spaces. Please, explain why this is relevant. Perhaps, it is possible to provide statements on the solution operators for some PDEs in wide use that are continuous operators between Sobolev spaces.

**The list of problematic statements, omissions, unclear parts, etc:**
1. [page 1] *"An example is an operator learning (Guo et al., 2016; Bhatnagar et al., 2019; Khooet al., 2021), which utilizes neural networks to parameterize the mapping from the parameters (external force, initial, and boundary condition) of the given PDE to the solutions of that PDE."* This description is misguided. The setting authors described is generic and not specific for neural operators.
2. [page 1] *"Another approach to operator learning is a neural operator, proposed in Li et al. (2020c;b;a). Li et al. (2020c) proposed an iterative architecture inspired by Green’s function of elliptic PDEs. The iterative architecture consists of a linear transformation, an integral operator, and a nonlinear activation function, allowing the architecture to approximate complex nonlinear mapping. An extension of this work, Li et al. (2020b) used a multi-pole method to develop a multi-scale graph structure. Gupta et al. (2021) approximated the kernel of the integral operator using the multiwavelet transform."* The description is significantly incomplete. There are dozens of neural operators, including DeepONet, wavelet neural operators, hierarchical attention, kernel-based methods, SVD-based neural operators, discretization-invariant convolutions, etc. Please, provide more examples or reference some review papers. In particular, I recommend the authors look at "HyperFNO: Improving the Generalization Behavior of Fourier Neural Operators."
3. [page 2] *"Figure 1: Comparison of the train and the test relative $L_2$ error by time horizon $t = 10, \dots, 19$ on the Navier-Stokes equation with viscosity $\nu = 10^{-5}$. FNO and MWT are highly overfitted, while PDNO is not."* Please, provide more details right here. What is a time interval used for training? What is a training strategy?  Is it an autoregression setting or a spatiotemporal "one-shot" prediction? What is the number of data points in train and test sets? etc
4. [page 2] *"Furthermore, the symbol may not be contained in a toroidal symbol class so that the Fourier integral operator cannot be guaranteed to be a continuous operator".* Please, clarify. It is easy to show that FNO is bounded in $L_2$ space by construction. It implies that it is a continuous operator.
5. [page 3] *"Analysis of Fourier integral operator based on Pseudo-differential operator".* This title is not accurate. The section does not contain an analysis of the Fourier integral operator. There are several definitions there, but the discussion of properties is absent.
6. [page 4, Definition 1] What is $\mathbb{T}^{n}$? Is it a torus? This notation is not defined.
7. [page 4] Section *"Proposed integral operator : Pseudo-differential integral operator".* The content of the section is simple but the presentation is poor. Please, explain the results to the reader in plain English. As I understand authors state that:

    i. In a continuous setting, there is a known relation between the smoothness of the symbol and the properties of pseudo-differential operator (references are given).

    ii. The known results are not directly applicable to the setting authors consider since they work with discrete Fourier transform, so the frequencies live on the lattice. Authors formulate and prove the result valid for the lattice (differential operator is replaced by difference operator).

    iii. Now, it is enough to show that chosen parametrization of the symbol (given by neural networks) fulfills the inequality used in the two statements above. The authors demonstrate that for GELU directly and provide more details in the appendix.

8. [page 4] *"The parameters $R(\xi)$ may not satisfy the condition of the toroidal symbol equation 10 in Definition 3 so that the Fourier integral operator cannot be guaranteed to be a continuous operator".* Forward reference to equation 10 unknown to the reader at a time. The claim about the discontinuity of FNO is questionable.
9. [page 7] *"Consider the general case which the input function $f_t(x)$ and the output function $f_{t+1}(x)$ are multi-valued function."* Probably, the authors mean a vector function, not an implicit (multi-valued) function.
10. [page 7] *"Toy example : 1D heat equation".* What is the purpose of this section? Perhaps, the authors adopt the approach from https://arxiv.org/abs/2003.03485, which contains a similar example but for an elliptic equation. What is the point of learning the symbol of the heat equation if we know in advance that PDNO has the same parametric form? It barely comes as a surprise that learning in this setting is possible.
11. [page 8] *"time grids $t = 0.05 + 0.1n$ ($n = 0, 1, \dots, 9$)"* Why time does not start from zero?
12. [page 8] *"dataset. For all the experiments, we use the PDNO that consists of four iterations of the network described".* Why iterations? "Layer" sounds more appropriate in this context.

**Additional experiments:**
1. The role of the smoothness of a symbol is not clear. In all scenarios, the authors consider infinitely smooth symbols. What if they are only continuous (e.g., neural networks with ReLU activations)? Is it going to lead to a wider generalization gap?
2. The fact that PDNO performs better than FNO is not enough to claim that the reason is the smoothness of PDNO in Banach space. One also needs to consider the problem with the non-smooth operator in Banach space. For example, a ReLU-like operator $$\\mathcal{A}[f] =  \\begin{cases} f,~ \\left \\| f \\right \\| \\geq a; \\\\ 0,~ \\left \\| f \\right \\| < a\end{cases}$$ for some positive real $a$. Is that true that for this operator FNO performs better than PDNO because PDNO constitutes unsuitable prior?
3. All experiments are performed on smooth inputs and outputs. What if both input and output are discontinuous? For example, simple advection equation with discontinuous data from https://arxiv.org/abs/2203.13181 will do.

**Strengths And Weaknesses:**

**Strengths**
1. Architecture modifications appear to be novel
2. Authors presented reasonable comparison with other architectures and performed ablation study

**Weaknesses** (details are given in the Requested Changes below)
1. Presentation

    Some parts of the article are not clearly explained, and the overall organization of the material can be improved.

2. Motivation

    It is not immediately clear why one needs to consider novel architecture. Which problem does it resolve and why? Similarly, the necessity of the introduced theory is questionable. Why should the readers care about pseudo-differential operators and toroidal symbols?

---

### Review · Reviewer_bQQP · 2023-10-01

**Summary Of Contributions:**

This paper develops a procedure to use the so-called pseudo differential integral operator (PDIO) to fit the solutions of certain partial differential equations. The idea is as follows. The Fourier integral operator is defined to be

K_R[f](x) = F^{-1}[R(xi) F[f](xi)](x)

where F is the Fourier transform and xi in Z are the discrete-valued frequencies. When used to obtain the solution, of say PDE L[u] = f for a linear operator L = sum_i c_i D^i, the solution u can be calculated using

a(xi) F[u] = F[f]

where a(xi) is the so-called “symbol”. The pseudo-differential operator generalizes the symbol to be a(x, xi), i.e., one can think of it as solution of the PDE using a local Fourier transform calculated at each point in the domain x. This is very well-studied and classical mathematics. There are two key ideas discussed in this paper:

* The general symbol a(x, xi) could be decomposed as a product over the problem domain and the Fourier frequencies as a_1(x) a_2(xi) to fit each term using a multi-layer perception (MLP);
* since the MLP-based approach will have to use the discrete Fourier transform when it performs its computations, the authors discuss a discretization of the Euclidean symbol (where xi is real-valued) into the “toroidal symbol” (where xi takes values in integers). There are existing results that suggest how to do this discretization effectively. The authors then discuss some minor variations of
* One can use pseudo-differential integral operators to build what is called a “neural operator” which approximates the solution of a PDE (or a map between two function spaces in general) layer-by-layer.

The experimental part of the paper demonstrates how to use this operator to calculate the solution of a heat equation, the porous medium equation with a constant boundary condition (Darcy flow), and the Navier Stokes equation for incompressible fluids. The authors argue that modeling the solution using the pseudo-differential neural operator enables the solver (i.e., the neural network) to predict the values of the solution accurately on a grid that is finer than the one in the training set that was used to fit the operator. The experimental results show that this approach has a smaller error than, say the Fourier neural operator, which does not depend upon x.

**Audience:**

Yes

**Claims And Evidence:**

Yes

**Requested Changes:**

Please see above. it would be useful to show detailed characterization of which problems this new neural operator would be well-suited for.

**Strengths And Weaknesses:**

This is a neat application of an existing mathematical concept that could be useful in solving PDEs that depend upon the state in a non-trivial fashion.

This is a very incremental paper that directly uses an existing idea. It is not clear for what kinds of problems this approach would perform better (the authors do show some results, but are these results conclusive...e.g., do we have enough evidence that this is indeed a replacement for a Fourier neural operator even for these equations in all cases?). In order for these results to be meaningful, there should be some argument, either theoretical or experimental, that characterizes the kinds of problems that will/will not be benefit from this approach, and why.

For problems that involve learning maps between function spaces, one should characterize the kinds of maps that can/cannot be learned via experiments a bit more precisely. It is not sufficient to simply say that a method obtains the solution for these few equations and so and so resolutions. Ideally, we would like to see detailed trends of how computational complexity of a different neural operator trades-off as the solution demands higher and higher resolutions. A simple example of this is the user trying out different Runge-Kutta methods, (i) one can always use a higher order method but it would be infeasible in any real problem with computational constraints, and (ii) the point of proposing a numerical method to solve a problem is to be able to crisply characterize the kinds of problems that the said method is well-suited for. Indeed, if a real-problem does not require a high-resolution solution of a PDE, then one would be perfectly happy with the Fourier neural operator.

The paper has a lot of grammatical errors. The authors are advised to proof read the narrative.

In the reviewers’s opinion, the discussion on the toroidal symbol is not directly relevant to the problem. Solutions to PDEs predicted by the learned neural operator will necessarily have a gap with respect to the true solution (due to sample sizes errors, incorrect modeling assumptions such as truncation of frequencies) and the discretization error of the symbol is just one among these. So in principle, while the argument of the authors that using the pseudo-differential integral operator is better than the Fourier operator is well-taken, it does not seem important to worry about the discretization of the PDIO.

---

### Decision · Action_Editor_8o5w · 2024-01-22

**Recommendation:** Accept with minor revision

**Comment:**

All reviewers are either accept or leaning toward accept, although notice incremental nature and misprints.

**Audience:**

Neural operators and ML for physics problems attracts more and more attention of machine learning community, which is confirmed by growing number of papers.

**Claims And Evidence:**

The paper proposes to use partial differential integral operator (PDIO) as a motivation for a new neural operator architecture.

The main claim is that compared to Fourier Neural Operator the PDIO architecture overfits less and shows favorable performance.